# Evaluation of Intraoperative Volumetric Assessment of Breast Volume Using 3D Handheld Stereo Photogrammetric Device

**DOI:** 10.3390/jpm13081262

**Published:** 2023-08-15

**Authors:** Rafael Loucas, Marios Loucas, Sebastian Leitsch, Katarina Danuser, Gabriela Reichard, Omar Haroon, Julius Michael Mayer, Konstantin Koban, Thomas Holzbach

**Affiliations:** 1Thurgau Hospital Group, Department of Hand and Plastic Surgery, CH-8500 Frauenfeld, Switzerland; marios.loucas@hotmail.com (M.L.); sebastian.leitsch@stgag.ch (S.L.); katarina.danuser@stgag.ch (K.D.); gabriela.reichard@stgag.ch (G.R.); omar@dromar.ch (O.H.); thomas.holzbach@stgag.ch (T.H.); 2Division of Plastic, Aesthetic and Reconstructive Surgery, Medical University of Graz, 8010 Graz, Austria; 3Department of Plastic and Hand Surgery, Inselspital, University Hospital Bern, CH-3012 Bern, Switzerland; juliusmichael.mayer@insel.ch; 4Division of Hand, Plastic and Aesthetic Surgery, University Hospital LMU, 81377 Munich, Germany; konstantin.koban@med.uni-muenchen.de

**Keywords:** intraoperative volumetric assessment, breast size, breast volume, VECTRA H2 handheld device, VECTRA, three-dimensional, 3-dimensional, intraoperative volume

## Abstract

Methods for assessing three-dimensional (3D) breast volume are becoming increasingly popular in breast surgery. However, the precision of intraoperative volumetric assessment is still unclear. Until now, only non-validated scanning systems have been used for intraoperative volumetric analyses. This study aimed to assess the feasibility, handling, and accuracy of a commercially available, validated, and portable device for intraoperative 3D volumetric evaluation. All patients who underwent breast surgery from 2020 to 2022 were identified from our institutional database. Intraoperative 3D volumetric assessments of 103 patients were included in this study. Standardized 3D volumetric measurements were obtained 3 months postoperatively to compare the intraoperatively generated volumetric assessment. All of the study participants were women with a mean age of 48.3 ± 14.7 years (range: 20–89). The mean time for intraoperative volumetric assessment was 8.7 ± 2.6 min. The postoperative 3D volumetric assessment, with a mean volume of 507.11 ± 206.29 cc, showed no significant difference from the intraoperative volumetric measurements of 504.24 ± 276.61 cc (*p* = 0.68). The mean absolute volume difference between the intraoperative simulations and postoperative results was 27.1 cc. Intraoperative 3D volumetric assessment using the VECTRA H2 imaging system seems to be a feasible, reliable, and accurate method for measuring breast volume. Based on this finding, we plan to investigate whether volumetric objective evaluations will help to improve breast symmetry in the future.

## 1. Introduction

Breast volume is a crucial factor and a key metric that defines breast symmetry, influencing cosmesis and patient satisfaction after breast surgery [1,2,3]. Several objective methods for evaluating breast volume and symmetry are widely used, and these methods are based on linear distances such as anthropometric measurements [4,5,6,7], volumetric measurements using medical imaging technology [8,9,10,11], techniques like water displacement [12,13,14,15,16], and casting techniques [17,18]. Three-dimensional (3D) photographic imaging measurements are increasingly gaining a role in breast surgery. Three-dimensional volumetric assessment enables an accurate and efficient analysis of breast volume and symmetry as opposed to traditional two-dimensional (2D) photography [1,19,20,21]. The use of 3D surface imaging has enabled quantitative linear distance measurements, surface and volumetric calculations of the breast region as well as objective breast symmetry evaluations between the breasts by virtually superimposing the mirrored breasts over each other [22,23]. Three-dimensional imaging systems can acquire spatial data through a 3D Cartesian coordinate system (x, y, z), and hence offer a unique advantage over 2D photography [24].

Precise preoperative and postoperative measurements are key elements for achieving postoperative breast symmetry [25,26]. Pre- and postoperative imaging are well-established tools in plastic surgery, specifically in breast surgery [2,27,28,29,30], and several studies have validated the use of imaging in volume measurement [29,30,31,32] as a surgical planning tool as well as a marketing tool in esthetic breast surgery. 3D images are used to illustrate to potential patients how they may appear after breast procedures such as augmentation or mammoplasty/mastopexy.

A recent survey investigating the current trends and controversies in breast augmentation showed that 15% of plastic surgeons in the USA use 3D-imaging systems in their practice [33]. Few of these systems are validated [2,34], and their use is primarily limited to preoperative and postoperative analyses with patients standing upright in front of the camera or scanning device. However, evaluating the intraoperative volume and symmetry is crucial for achieving an optimal esthetic result [35]. Intraoperative assessment of breast volume and symmetry mainly relies on the surgeon’s visual estimations, which can be influenced by individual subjective factors [28], and as such may result in bias. To the best of our knowledge, only one study has evaluated the use of intraoperative 3D volumetric assessment to date. Yang et al. [35] conducted a study on an intraoperative objective methodology that utilized a handheld 3D scanner to assist in symmetrical assessment during vertical reduction mammaplasty with a superior pedicle. The authors showed that intraoperative 3D scanning provided a reliable method to aid in symmetry adjustments and ensure improved postoperative breast symmetry. However, the utilized system was not validated. Measurements were taken with the patient lying in the supine position. In addition, postoperative analyses were performed using a different method. Moreover, three-dimensional surface imaging systems could not be used intraoperatively due to several reasons such as the inability of automatic rendering by a draped patient, the operating position, etc. 

The VECTRA H2 3D imaging system (Canfield Scientific^®^, Parsippany-Troy Hills, NJ, USA) is a commercially available 3D imaging system, and has been validated for preoperative and postoperative volumetric measurements [2,34]. VECTRA H2 3D technology (Canfield Scientific^®^, Parsippany-Troy Hills, NJ, USA) uses 3D symmetry assessment. The software measures breast symmetry by recording digital photographs of the patient’s right and left breasts in multiple views individually, then assesses the differences in symmetry by overlaying the corresponding views of the right and left breast onto each other [34]. The VECTRA H2 3D software (version 5.7.2)calculates the average distance between the two breast surfaces and produces a number reflective of breast symmetry [36]. Additionally, the VECTRA^®^ technology assesses the breast as a 3D structure, considering breast volume or projection data in comparison with the 2D photography [2,34,34,36]. The breast volume is subsequently computed using the distance between the breast surface and this virtual chest wall [2,34,36]. This is a convenient and inexpensive method for calculating breast volume, and it has been shown to have an average accuracy of about a 2.2% underestimation of the true breast volume (range −2.17 to −2.28%) [34]. The VECTRA three-dimensional surface imaging of the breasts for measuring pre- and postoperatively breast volume and shape symmetry has been validated in several studies [2,34,36,37]. Clinicians and researchers suggest that measurements using the VECTRA technology three-dimensional surface imaging system have the potential to assist in preoperative planning and also as a measure of esthetic outcome [2,34,37].

Recent advancements in surgical technology have emphasized the importance of intraoperative measurements for achieving the optimal esthetic outcomes in breast surgery [35]. While pre- and postoperative imaging have proven valuable in planning and evaluating surgical results, the need for real-time, objective, and validated intraoperative volumetric assessments remains unmet [2,34]. Intraoperative adjustments to breast volume and symmetry significantly impact the final outcome, however, the current methods rely largely on subjective visual estimations [28]. 

Thus, the use of intraoperative volumetric measurements in this population remains unknown. Addressing this gap, our study aimed to investigate the feasibility, handling, and accuracy for intraoperative volumetric measurements using the VECTRA H2 3D imaging system (Canfield Scientific^®^, Parsippany-Troy Hills, NJ, USA), a popular 3D imaging system used worldwide. By evaluating the system’s potential to provide reliable real-time data during surgery, we seek to enhance the precision and success of breast surgical procedures and improve patient satisfaction.

## 2. Materials and Methods

### 2.1. Study Design and Study Population

This was a retrospective study that evaluated the volumetric assessment outcomes of patients who underwent breast surgery including reduction mammaplasty with superomedial pedicle, mastopexy, and breast augmentation. The study was approved by our institutional research committee (Spital Thurgau HPC Research Committee; 6 August 2020; AZ:HPC2020-3). No additional approval of the Kanthonal Ethics Board (EKOS Ostschweiz) was sought for following reasons: Photographic Data Analysis was performed retrospectively on anonymised patient data, no additional examination of patients was planned/performed, no potential change of treatment was implied. The study was performed in accordance with the ethical standards as laid down in the 1964 Declaration of Helsinki and its later amendments.

### 2.2. Patient Selection

Between January 2020 and March 2022, our institutional breast database meticulously recorded 688 consecutive breast procedures, encompassing reduction mammoplasty, mastopexy, and augmentation mastopexy. For the study, strict eligibility criteria were applied to select suitable participants. Specifically, patients were required to have undergone primary breast surgery, completed both clinical and volumetric follow-ups, and provided informed consent to participate. As a result of these rigorous criteria, 186 patients were excluded from the study. Among the exclusions, 49 patients lacked complete clinical and volumetric follow-up, while 137 patients did not have access to intraoperative three-dimensional (3D) volumetric measurements. Furthermore, to maintain a focused and homogenous study group, certain types of breast surgeries were excluded from the analysis. These exclusions included revision breast surgeries (*n =* 21), transgender surgeries (*n =* 1), gynecomastia operations (*n =* 19), and breast lipofilling procedures for one or both breasts (*n =* 49). Following the careful application of these criteria and exclusions, the final cohort consisted of 103 patients who met all of the inclusion criteria for the study, as visually represented in Figure 1.

### 2.3. Surgical Technique

All procedures were performed by four senior, board-certified plastic surgeons in our department. The operations were performed in a standardized manner: antibiotic prophylaxis was administered intravenously 30 min before skin incision using 1.5 g Cefuroxime (Fresenius Kabi, Kriens, Switzerland). General anesthesia was used in all cases.

The conducted procedures were performed as follows. Reduction mammoplasties were exclusively performed using the superomedial pedicle technique. Mastopexies were undertaken employing both the wise pattern and vertical methods. Implant-based breast augmentation procedures encompassed meticulous subglandular and dual-plane pocket preparation techniques. Augmentation mastopexies were solely executed utilizing the wise pattern mastopexy approach, in conjunction with either subglandular or dual-plane pocket preparation.

### 2.4. Three-Dimensional (3D) Volumetric Assessment

Intraoperatively, all patients were positioned in an upright beach chair position before skin closure. For the three-dimensional (3D) volumetric assessment, we utilized the VECTRA H2 3D imaging system (Canfield Scientific^®^, Parsippany-Troy Hills, NJ, USA). At our institution, intraoperative volumetric assessments are routinely conducted as part of clinical procedures.

The device above-mentioned is a handheld 3D camera (400 g) used with stereo photogrammetry technology for 3D imaging manufactured by Canfield Scientific^®^ (Parsippany-Troy Hills, NJ, USA). The 3D camera uses infrared sensors to capture depth information. Using depth information and a built-in camera, the VECTRA H2 3D imaging system can capture objects in three dimensions at a rate of up to 30 frames per second. The utilized system has a shutter speed of two milliseconds (ms) and consists of dual light point positioning assistance to ensure the correct distance between the camera system and the object. The captured volume in this specification is 700 mm (height) × 410 mm (with) × 400 mm (depth). A non-sterile individual was responsible for taking photographs and conducting 3D analyses in the operating room (OR). Photographic images were captured using the VECTRA H2 device from three different positions (Figure 2). Three images were taken: two from a lateral angle (one from the right and one from the left at a 45° angle with the target point in the center of the corresponding inframammary fold) and one from a central angle (with the target point located substernally).

For 3D analysis, the device was connected to a notebook computer to calculate the 3D model using the VECTRA volumetric 3D surface imaging system software (Canfield Scientific, Parsippany-Troy Hills, NJ, USA) (Figure 3). The three intraoperatively taken images were imported into the VECTRA volumetric 3D surface imaging system software, and specific landmarks were automatically detected by the VECTRA^®^ software: jugular notch, mid-clavicular, upper edge of the nipple–areolar complex (NAC), center of nipple, and medial and lateral aspects of the infra-mammary fold. Where automatic landmark detection was unsuccessful, the landmarks were manually placed or adjusted. The software generated a virtual thoracic wall using the contours of the patient’s 3D photographs and then extrapolated the volume of the overlying breast tissue. The breast volume was then measured in cubic centimeters (Figure 3).

### 2.5. Postoperative Three-Dimensional (3D) Volumetric Assessment

At three months postoperative, routine volumetric analysis via VECTRA volumetric 3D surface imaging system software (Canfield Scientific, Parsippany-Troy Hills, NJ, USA) was performed during consultation. In addition, standard clinical assessment, outcome measures, and conventional photographic images were taken.

### 2.6. Data Collection, Statistical Analysis, and Literature Review

Patient data were collected anonymously using the ELO software/electronic data capture system (ELO Digital Office GmbH, Stuttgart, Germany). All statistical analyses were conducted using IBM SPSS Statistics software, version 28.0 (Chicago, IL, USA). The normal distribution of variables was assessed using the Shapiro–Wilk test. Pre- and postoperative scores were compared using either the paired t-test for parametric data or the Wilcoxon rank-sum test for nonparametric data. Fisher’s exact test was used to analyze the categorical variables, and a *p*-value of ≤0.05 was considered statistically significant.

## 3. Results

### 3.1. Patients and Demographics

One hundred and three (103) patients, all female, underwent breast surgery with intraoperative volumetric assessment using the three-dimensional (3D) technology, and these patients were included in the analysis. The mean age of the patients was 48.34 ± 14.7 years (range: 20–89); the mean body weight was 68.31 kg (range: 45–164 kg); the patients’ mean body height was 164.42 cm (range: 90.2–182 cm); the mean body mass index (BMI) was 24.65 (range: 16.52–38.26) (Table 1).

### 3.2. Volumetric Outcomes

The mean time for intraoperative volumetric assessment including image analysis was 8.7 ± 2.6 min (95% CI: 6.5 min to 13.1 min). The three-month postoperative three-dimensional (3D) volumetric assessment showed a mean volume of 507.11 ± 206.29 cc, while the mean volume during intraoperative volumetric measurement was 504.24 ± 276.61 cc. No significant differences were found between the intraoperative and three-month postoperative volumes (*p* = 0.68). The mean absolute volume difference between the intraoperative simulations and postoperative results was 27.1 cc (range: 0.9–83.5 cc; standard deviation [SD] = 20.7 cc) (Table 2).

## 4. Discussion

This study demonstrates the feasibility, reliability, and accuracy of using the VECTRA H2 3D imaging system for the intraoperative three-dimensional (3D) volumetric assessment of breast volume. The time required for intraoperative volumetric assessment—including image analysis—with a mean of 8.7 min, is suitable for use in intraoperative settings.

It is important to emphasize that these values were compared to the values measured three months postoperatively. We are aware that breast appearance can continue to change after three months. However, we decided to use this time point because any inaccuracies caused by swelling should have subsided after three months. Additionally, ptosis, or bottoming out—which can cause further inaccuracies in 3D breast volumetric measurements—has not yet occurred [35,38,39]. We chose this time point based on data from the studies by Eder et al. [40] and Creasmen et al. [41]. Both studies showed that there was no significant difference in breast volume and contour between the three-month and six-month postoperative periods. 

Precise preoperative markings and appropriate adherence to them during surgery used to be the key elements for achieving postoperative breast volume and symmetry [7,15,16,25,26]. However, evaluating the intraoperative volume seems to play a crucial role in achieving the best possible postoperative aesthetic results [35]. So far, the intraoperative assessment of breast volume and volume differences mainly relies on the surgeon’s visual estimations, which can be influenced by individual subjective factors [15,16]. Therefore, it is important to use an objective tool such as intraoperative 3D volumetric assessment to ensure symmetrical breast volume.

As the field of breast surgery continues to evolve, the quest for improved precision and patient satisfaction remains paramount. In this pursuit, the utilization of advanced 3D imaging systems like the VECTRA H2 holds great promise [25,26]. Not only does it provide a valuable tool for preoperative planning and postoperative assessment, but its potential for intraoperative use could revolutionize the way breast volume and symmetry adjustments are carried out during surgery. By offering a more objective and accurate means of measuring breast volume and symmetry in real-time, the VECTRA H2 3D imaging system has the potential to enhance surgical outcomes and optimize the esthetic results [2,34,37]. 

Additionally, the integration of 3D imaging technology into breast surgery practices has the added advantage of facilitating better communication between plastic surgeons and their patients [2,34]. With the ability to visualize and demonstrate potential outcomes in a three-dimensional format, patients can gain a clearer understanding of the proposed surgical changes, enabling them to make more informed decisions about their procedures [2,34,36,37]. This enhanced communication can lead to increased patient satisfaction and confidence in the surgical process. Moreover, the objective data provided by the VECTRA H2 3D imaging system can foster greater transparency and trust between surgeons and patients, reinforcing the importance of evidence-based approaches in the field of esthetic breast surgery. With these potential benefits in mind, exploring the feasibility of intraoperative volumetric measurements using the VECTRA H2 3D imaging system becomes an exciting and vital avenue for advancing the art and science of breast surgery. However, care must be taken not to evoke unrealistic expectations. As much as we appreciate the additional information obtained from the preoperative and especially intraoperative volumetric analyses, we never hand out 3D simulations to our patients. We acknowledge the opportunity these systems give us to simulate a possible postoperative result, but we take care to only use these simulations to better understand the patients’ expectations and not to create unrealistic images, which may be mistaken for guaranteed results.

The VECTRA H2 3D imaging system can also be used very effectively in breast augmentation surgery, especially in cases with breast asymmetries and asymmetric reduction surgeries. This could be helpful for a more refined selection of implant size in cases of asymmetrical breasts. The implants are usually pre-selected preoperatively, sometimes already today based on 3D volumetric analyses. Nevertheless, in many cases intraoperatively, the results are not satisfactory, so surgeons switch to different implants. However, intraoperative decision-making is highly subjective, relies on the surgeon’s visual appraisal, and therefore corresponds to their experience. It would be great to have an objective tool to support intraoperative decision-making. The same applies to breast reduction cases. In our experience, most large and ptotic breasts also present a volumetric asymmetry to some degree. Preoperatively, it is relatively easy to detect which breast is larger or more ptotic. While linear differences can be easily measured intraoperatively, it is much harder to analyze possible residual volumetric differences after tissue resection. Hopefully, intraoperative 3D volumetric measurements will allow for more precise and tailored intraoperative adjustments like additional tissue resection or fat grafting before skin closure. This study shall lay the groundwork for further investigations.

Several studies have compared 2D and 3D scanning techniques and have shown that using 3D scanning for objective breast volume and symmetry evaluation is easy, sufficiently fast, observer-independent, and more precise than 2D measurements [19,20,21,34,42]. While the Breast Analyzing Tool (BAT^®^) enables the objective evaluation of breast symmetry by generating a breast symmetry index (BSI) and demonstrates a high correlation between subjective and objective BSI values, quantifiable volumetric information is not obtained [43]. Nevertheless, a detailed comparison of that method and the 3D volumetric measurements might be interesting for future investigation. Eder et al. developed a 3D evaluation protocol to analyze breast symmetry by comparing the 3D surface imaging of the left and right breasts. They evaluated the potential clinical application of this protocol by comparing it to the established 2D BCCT core software [42,44,45]. The authors demonstrated that the evaluation of breast symmetry in 3D is independent of observer bias and significantly more precise than a 2D evaluation [42].

In this study, intraoperative 3D volumetric assessment, which involved image analysis, took 8.7 min. However, there was a steep learning curve when it came to correctly analyzing the images and setting landmarks. The software used, VECTRA volumetric 3D surface imaging system software (Canfield Scientific, New Jersey, USA), allows for automatic landmark placement and image calibration through the use of a standard gauge within the image plane. The system’s precision is very high, enabling a 3D resolution based on a triangle edge length of 1.2 mm, in body scans 3.5 mm, respectively. However, intraoperatively, one can only obtain the volumetric measurement of each individual breast in total. By analyzing the generated images on screen, the surgeon may detect possible differences between the quadrants of the breast of either side, but no separate analysis of each quadrant of the breast can be generated automatically. The system allows for image mirroring of both breasts including overlapping and color marking of the volumetric differences, but this requires time and technical skills.

To the best of our knowledge, this is the first study to establish an intraoperative 3D volumetric analysis using a validated system [2,34,37,46]. In addition, this study comprised the largest sample size of all intraoperative 3D volumetric studies in breast surgery. Nevertheless, despite the clear benefits of intraoperative 3D volumetric assessment, we acknowledge certain limitations and technical challenges in this study that still need to be addressed. One limitation of this study is the positioning of the patients intraoperatively. The system was validated in a scenario where the patient was standing upright in front of the camera. Intraoperatively, we bring the patients into an upright sitting position. Our findings strongly suggest that this position is adequate to obtain a valid volumetric result, but this aspect must be analyzed in greater detail in the future. Although the patients’ data were prospectively enrolled in our institutional breast database, this study does not fulfill all the criteria of a prospective design. Although the cost of 3D imaging devices has significantly decreased in recent years, the total cost for an overall 3D solution (soft- and hardware) is still around USD 20,000–30,000 [42]. Furthermore, although we conducted preoperative assessments on every patient using 3D planning software and meticulous objective anthropometric measurements, we did not utilize dedicated 2D planning methods such as MRI scans or mammograms. Moreover, we are aware of the time consumed during the intraoperative scanning and image analysis. 

Another possible source of inaccuracy in the 3D volumetric assessment, in general, is the dorsal boundary of the breast generated by the virtual chest wall. Using the 3D imaging system software, the dorsal limits were not visible, and the body surface surrounding the breast had an influence on the interpolation of the dorsal limit, which is used for the volume calculation. Thus, small patient thoracic deformity or rotation whilst obtaining the 3D photographs can affect the accuracy of the measurements. The form of the thorax itself or the amount of subcutaneous fatty tissue, for example, could have a relevant influence on the interpolation. However, the user can manually correct small rotations using the computer software to align the xyz coordinates. To minimize the effect of rotation, anatomical landmarks were marked prior to photography and the patients were positioned in an upright beach chair position. We decided not to include preoperative volumetric measurements into our study. We have worked with this system for many years to generate preoperative analyses and assess postoperative results. Unfortunately, preoperative 3D volumetric assessments do not yield valid results in ptotic breasts. The reason is that the virtual thoracic wall cannot be generated correctly if the breast is covering the inframammary fold.

We believe that 3D imaging systems for the intraoperative 3D volumetric assessment of breast volume will play an important role in the evaluation of breast symmetry in the future, thus improving cosmetic outcomes in breast surgical procedures. As the technology evolves, 3D imaging systems could be used as a standard intraoperative tool in breast surgery.

## 5. Conclusions

Breast symmetry is one of the dominant indicators of overall esthetic outcome. Intraoperative three-dimensional (3D) volumetric assessment using the VECTRA H2 3D imaging system appears to be a feasible, reliable, and accurate method for measuring breast volume. Based on this finding, we plan to investigate whether using volumetric objective evaluations will improve breast symmetry in the future. This study is unique in measuring the breast volume intraoperatively in a large population using a 3D imaging system that is commercially available and validated for preoperative and postoperative volumetric measurements. 

## Figures and Tables

**Figure 1 jpm-13-01262-f001:**
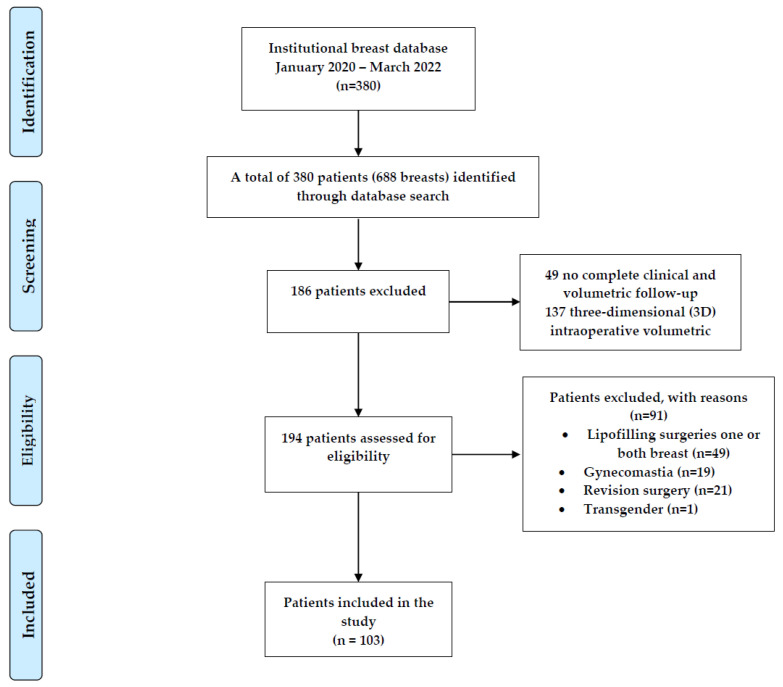
Flowchart of the institutional database and patient identification.

**Figure 2 jpm-13-01262-f002:**
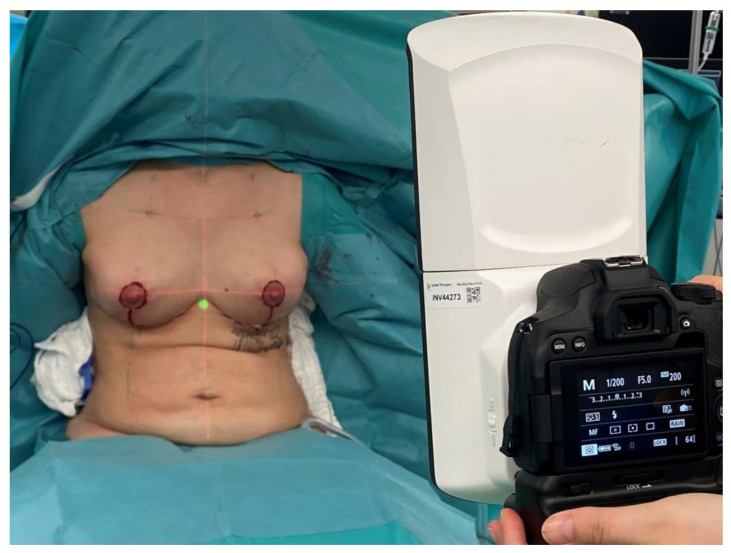
Photographic images using the VECTRA H2 device were taken from three different positions. The patient’s correct upright position and symmetrical nipple height were verified by laser level projection (horizontal and vertical red projection lines).

**Figure 3 jpm-13-01262-f003:**
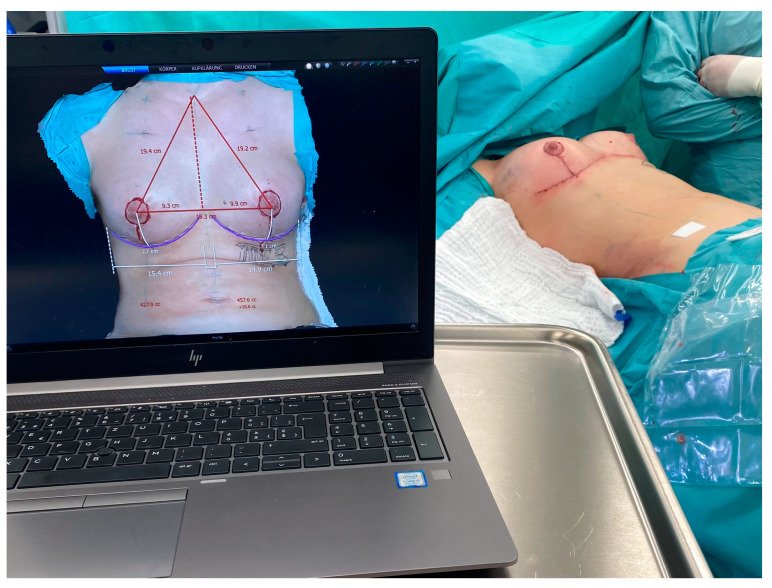
Intraoperative volumetric assessment was executed via VECTRA volumetric 3D surface imaging system software (Canfield Scientific, Parsippany-Troy Hills, NJ, USA). The total duration for photographic capturing, transfer to a notebook computer, and image analysis was 8.7 ± 2.6 min (95% CI: 6.5 min to 13.1 min).

**Table 1 jpm-13-01262-t001:** Demographic data for the study groups.

	Patients Mean ± Std (Min, Max)
Age at surgery	48.34 ± 14.73 (20; 89), *n* = 103
Gender	103 Female
Side (Uni/Bilateral)	103 Bilateral
Body Mass Index	24.65 ± 3.88 (16.52; 38.26), *n* = 103
Diabetes	6/103

**Table 2 jpm-13-01262-t002:** Intra- and postoperative three-dimensional (3D) volumetric measurements of the 103 patients available for clinical and volumetric follow-up. Values are given as mean ± SD [95% confidence interval] or (minimum; maximum).

	Patients	*p* Value
Intraoperative three-dimensional (3D) volumetric measurement	507.11 ± 206.29 cc *n* = 103	-
Postoperative three-dimensional (3D) volumetric measurement	504.24 ± 276.61 cc *n* = 103	-
Mean absolute volume difference between intraoperative simulations and postoperative results	27.1 ± 20.7 (0.9; 83.5) *n* = 103	0.68

## Data Availability

Supporting data are available from the authors upon request.

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
