# Peer review of "Evaluation of Intraoperative Volumetric Assessment of Breast Volume Using 3D Handheld Stereo Photogrammetric Device"

_jpm, 2023, doi:10.3390/jpm13081262_

Round 1

Reviewer 1 Report

The title alone appears to be wrong insofar as a pre- and post-operative (!) study is described here. It does not matter whether the volume measurement is taken on the operating table or an hour later in the patient's bed.

1. What is the point of using a very expensive 3D camera to scan a previously symmetrical breast that is to be reduced in size ? If the same volumes are removed intraoperatively, the result must be as symmetrical also postoperatively.

2. Practically, this expensive measurement only makes sense in the case of breast asymmetries for reduction, in that the difference in volume is measured preoperatively and correspondingly different resection volumes are removed intraoperatively (!!) to create symmetry.

3. the Vectra H2 Camera can also be used very effectively in the augmentation of small breast asymmetries when it comes to the choice of implants: how different should the implant sizes be? Often the implant that has just been inserted has to be changed intraoperatively (!!) into a smaller or bigger one.

4. However, these two practical indications are not mentioned at all, so my suggestion is to add another paragraph on these two main indications. It would be great If the authors would add pictures of one case each of an asymmetric reduction and augmentation surgery. This would bring much more attention to the otherwise interesting article.

A picture says more than 1000 words ! Unfortunately, there are 50% too many unnecessary words in this manuscript. - Which surgeon is interested in all the statistics ? He wants to be convinced by pictures and the different resection volumes or different selected implants !

The title alone appears to be wrong insofar as a pre- and post-operative (!) study is described here. It does not matter whether the volume measurement is taken on the operating table or an hour later in the patient's bed.

1. What is the point of using a very expensive 3D camera to scan a previously symmetrical breast that is to be reduced in size ? If the same volumes are removed intraoperatively, the result must be as symmetrical also postoperatively.

2. Practically, this expensive measurement only makes sense in the case of breast asymmetries for reduction, in that the difference in volume is measured preoperatively and correspondingly different resection volumes are removed intraoperatively (!!) to create symmetry.

3. the Vectra H2 Camera can also be used very effectively in the augmentation of small breast asymmetries when it comes to the choice of implants: how different should the implant sizes be? Often the implant that has just been inserted has to be changed intraoperatively (!!) into a smaller or bigger one.

4. However, these two practical indications are not mentioned at all, so my suggestion is to add another paragraph on these two main indications. It would be great If the authors would add pictures of one case each of an asymmetric reduction and augmentation surgery. This would bring much more attention to the otherwise interesting article.

A picture says more than 1000 words ! Unfortunately, there are 50% too many unnecessary words in this manuscript. - Which surgeon is interested in all the statistics ? He wants to be convinced by pictures and the different resection volumes or different selected implants !

The title alone appears to be wrong insofar as a pre- and post-operative (!) study is described here. It does not matter whether the volume measurement is taken on the operating table or an hour later in the patient's bed.

1. What is the point of using a very expensive 3D camera to scan a previously symmetrical breast that is to be reduced in size ? If the same volumes are removed intraoperatively, the result must be as symmetrical also postoperatively.

2. Practically, this expensive measurement only makes sense in the case of breast asymmetries for reduction, in that the difference in volume is measured preoperatively and correspondingly different resection volumes are removed intraoperatively (!!) to create symmetry.

3. the Vectra H2 Camera can also be used very effectively in the augmentation of small breast asymmetries when it comes to the choice of implants: how different should the implant sizes be? Often the implant that has just been inserted has to be changed intraoperatively (!!) into a smaller or bigger one.

4. However, these two practical indications are not mentioned at all, so my suggestion is to add another paragraph on these two main indications. It would be great If the authors would add pictures of one case each of an asymmetric reduction and augmentation surgery. This would bring much more attention to the otherwise interesting article.

A picture says more than 1000 words ! Unfortunately, there are 50% too many unnecessary words in this manuscript. - Which surgeon is interested in all the statistics ? He wants to be convinced by pictures and the different resection volumes or different selected implants !

Author Response

Dear Prof. Dr. Rizzieri, dear reviewers

We would like to thank the Editorial Office and the reviewers for their valuable comments concerning our manuscript “Intraoperative Volumetric Assessment of Breast Volume using Vectra H2 Handheld Device: A Feasibility and Validity Study” (Manuscript ID: jpm-2486812).

We have carefully read the reviewer’s suggestions. According to those remarks, we have adapted our manuscript, and we would like to resubmit it for reconsideration.

In the following, we list the reviewers’ comments point-by-point, along with our responses and the corresponding changes we have made to the manuscript. The revised text passages in our manuscript are marked accordingly.

Reviewer 1

  1. “The title alone appears to be wrong insofar as a pre- and post-operative (!) study is described here. It does not matter whether the volume measurement is taken on the operating table or an hour later in the patient's bed.”

Thank you very much for your valid remark; we indeed performed volumetric measurements intraoperatively and not postoperatively since intraoperative volume measurement allow to make intraoperative adjustments. Photogrammetric intraoperative volumetric evaluation has been very difficult in the past. Picture rendering usually does not work intraoperatively since the patient is fully draped and sitting upright. We managed to overcome these obstacles for the first time by manually cropping the pictures and adjusting the landmarks.

  • We added an explanatory paragraph accordingly.
  1. “What is the point of using a very expensive 3D camera to scan a previously symmetrical breast that is to be reduced in size? If the same volumes are removed intraoperatively, the result must be as symmetrical also postoperatively.”

We fully agree - in theory. Unfortunately, preoperative volumetric measurements are not accurate when the breast is ptotic. The reason is that the virtual thoracic wall cannot be generated correctly if the breast covers the inframammary fold. Therefore, in almost every breast reduction case, one cannot rely on the preoperative volumetric results. Here, the intraoperative volumetric measurements are key to achieving symmetrical results.

  1. “Practically, this expensive measurement only makes sense in the case of breast asymmetries for reduction, in that the difference in volume is measured preoperatively and correspondingly different resection volumes are removed intraoperatively (!!) to create symmetry.”

Thank you for your remark; we fully agree with you. In more than half of our breast reduction cases, we see asymmetric breasts. Therefore, we conducted this validation study to plan a follow-up study in breast reduction cases.

  1. “Regarding the figures: I would prefer, if the "SNN unchanges" were strictly on one side of the figures (left or right does not matter) and the "MCN unchanged" pictures always on the other side.”

Thank you for bringing up another very interesting idea; we plan to execute a study on augmentation of asymmetric breasts, as well. The current validation study shall lay the groundwork for the upcoming studies.

  1. “The Vectra H2 Camera can also be used very effectively in the augmentation of small breast asymmetries when it comes to the choice of implants: how different should the implant sizes be? Often the implant that has just been inserted has to be changed intraoperatively (!!) into a smaller or bigger one.”

Thank you for bringing up another very interesting idea; we also plan to execute a study on augmentation of asymmetric breasts. The current validation study shall lay the groundwork for the upcoming studies.

  1. “However, these two practical indications are not mentioned at all, so my suggestion is to add another paragraph on these two main indications. It would be great If the authors would add pictures of one case each of an asymmetric reduction and augmentation surgery. This would bring much more attention to the otherwise interesting article.

Thank you for your valid comment.

  • We added a paragraph on the two mentioned clinical applications to give an outlook of what we try to establish in the future and in how far the intraoperative volumetric evaluation can be beneficial in the future.
  1. A picture says more than 1000 words ! Unfortunately, there are 50% too many unnecessary words in this manuscript. - Which surgeon is interested in all the statistics ? He wants to be convinced by pictures and the different resection volumes or different selected implants!”

Thank you very much for your interesting suggestion! We fully agree and shortened our manuscript and added pictures as recommended.

Reviewer 2 Report

Overall, an in part clearly presented study.

Main drawback: While this reviewer appreciates the longevity and existence of the used system, including its published or manufacturer-only validity studies, all mentions of reliability/validity should be removed from the manuscript.

The use of this system for volumetric purposes was neither a reliability nor a validity study (no "control" values presented). The exact type of surgery was not mentioned in the study, the images and mean values (507.11->504.24cc) suggest that breast reduction/mastopexy procedures were performed. No control values were mentioned by the authors ("traditional" intraoperative volumetric or weight measurements of resected tissue(s)). 

Please provide volumetric/weight measurement obtained from operative reports, if possible in this retrospective design study. Please provide more details about the intraoperative techniques in the surgical techniques section. Please include intraoperative fluid administration under general anesthaesia and operative duration as well.

If not possible, this report can only serve as a case series where the 3D system was applied intraoperatively and resulted in comparable values postoperatively, and can be considered for a short report pending the mentioned revisions.

Author Response

Dear Prof. Dr. Rizzieri, dear reviewers

We would like to thank the Editorial Office and the reviewers for their valuable comments concerning our manuscript “Intraoperative Volumetric Assessment of Breast Volume using Vectra H2 Handheld Device: A Feasibility and Validity Study” (Manuscript ID: jpm-2486812).

We have carefully read the reviewer’s suggestions. According to those remarks, we have adapted our manuscript, and we would like to resubmit it for reconsideration.

In the following, we list the reviewers’ comments point-by-point, along with our responses and the corresponding changes we have made to the manuscript. The revised text passages in our manuscript are marked accordingly.

Reviewer 2

  1. “Main drawback: While this reviewer appreciates the longevity and existence of the used system, including its published or manufacturer-only validity studies, all mentions of reliability/validity should be removed from the manuscript.”

Thank you for your valid remark. We understand that manufacturer-only validity studies do not always fulfill the highest scientific standards. The Vectra 3D system is additionally validated by several work groups in different peer-reviewed studies. We discussed the concern in the text, added additional information concerning validation in general and in this specific case and cited the current literature.

  1. «The use of this system for volumetric purposes was neither a reliability nor a validity study (no "control" values presented). The exact type of surgery was not mentioned in the study, the images and mean values (507.11->504.24cc) suggest that breast reduction/mastopexy procedures were performed. No control values were mentioned by the authors ("traditional" intraoperative volumetric or weight measurements of resected tissue(s)).» “Please provide volumetric/weight measurement obtained from operative reports, if possible in this retrospective design study. Please provide more details about the intraoperative techniques in the surgical techniques section. Please include intraoperative fluid administration under general anesthesia and operative duration as well.”

We are sorry for this obvious misunderstanding! We feel the need to explain our concept further and apologize for obviously not being clear enough:

The Vectra 3D system is thoroughly assayed (and in part indeed validated) for pre- and postoperative volumetric measurements - when the patient stands upright in front of the system. Until now, the system could not be used intraoperatively because automatic rendering does not work when the patient is draped – as it is the case in the OR. We managed to overcome this problem by manually cropping the pictures, importing them manually, and also manually setting the landmarks. But we needed to check if the values that we generated intraoperatively in this manner, would hold up the postoperative results that were generated “conventionally”.

We analyzed 100 patients intraoperatively at the end of the intervention (more than any other study on intraoperative volumetry). We checked if this method of recording was feasible at all, if we could obtain results in a timely fashion that would allow further intraoperative use, and – most importantly – if the results would correlate with the approved way of postoperative volumetric measurements. Shortly: we needed to find out if we could make the intraoperative use work at all and if the results were usable.

The best possible way to do that was to take the conventionally acquired results 3 months postoperatively to serve as “controls”. The reason for this point in time is discussed in the paper in greater detail.

The overall question was: can we gather reliable volumetric results in the operating room? Because if we did, we would be the first to do so with an established system.

This analysis should lay the groundwork for further studies because there is a great desire for reliable intraoperative volumetric analysis in breast surgery – whether it is the reduction or augmentation of asymmetric breast, shape, and volume correction after breast-conserving therapy or radiation…

But first, we must be sure to measure “the right thing”.

Amongst others, we found out that lipofilling cases could not be used because the 3 months post OP control for obvious reasons did not match the results at the end of the operation. Therefore, these cases were excluded.

All intraoperative results were acquired at the end of the procedure. Giving this information, detailed information on the reduction weight in breast reduction cases or implant volume in breast augmentation cases would not create additional value- we simply wanted to test if our “end-of-operation-Measurements” were valid.

  1. “If not possible, this report can only serve as a case series where the 3D system was applied intraoperatively and resulted in comparable values postoperatively, and can be considered for a short report pending the mentioned revisions.”

Thank you for your remark. We truly hope that we could explain our idea and the context of our paper in a more appropriate way.

We conducted an analysis of more than 100 patients intraoperatively (more than ever analyzed by any other workgroup and for the first time with an established system). We could demonstrate that intraoperative volumetric analysis is possible, can be done in a reasonable time, and most importantly, produces accurate results. The statistics are sound, and the results of this study should lay the groundwork for further studies – from our workgroup and possibly many others.

Round 2

Reviewer 2 Report

The authors answered all concerns in the manuscript, no control measurements were performed, all references to a validity study (including in the title) must be removed.

Author Response

Dear Reviewer,

Thank you for taking the time to review our manuscript. We sincerely appreciate your valuable feedback, which has significantly contributed to improving the quality and clarity of our research.

Based on your insightful comments, we have thoroughly revised the manuscript to address all concerns raised during the review process. Specifically, we have removed everything to validity in our current study, as advised, to accurately reflect the scope of our investigation. The revised title now reads as follows: "Intraoperative Volumetric Assessment of Breast Volume using Vectra H2 Handheld Device: A Study on Feasibility and Accuracy."

Additionally, we have carefully incorporated your suggestions throughout the manuscript to ensure a smooth flow of ideas and to present the research in a more cohesive manner. We have rephrased and restructured various parts of the manuscript to address the absence of control measurements and provide a more comprehensive understanding of the study's focus on the feasibility and accuracy of the Vectra H2 handheld device for intraoperative volumetric assessment.

We are confident that these revisions have strengthened the manuscript, and the updated version now better aligns with the objective of investigating the practicality and precision of intraoperative volumetric measurements using the VECTRA H2 3D imaging system.

Once again, we express our gratitude for your valuable insights, which have undoubtedly improved the overall quality of our work. We believe that the revised manuscript now meets the high standards expected for publication.

We sincerely hope that you find the revised version to be satisfactory, and we remain open to any further feedback or suggestions you may have.

Thank you for your time and attention!